# Slow and Fast-Growing Chickens Use Different Antioxidant Pathways to Maintain Their Redox Balance during Postnatal Growth

**DOI:** 10.3390/ani13071160

**Published:** 2023-03-25

**Authors:** Edouard Coudert, Elisabeth Baéza, Pascal Chartrin, Justine Jimenez, Estelle Cailleau-Audouin, Thierry Bordeau, Cécile Berri

**Affiliations:** Institut National de Recherche pour l’Agriculture, l’Alimentation et l’Environnement, Université de Tours, UMR BOA, F-37380 Nouzilly, France

**Keywords:** chicken, antioxidant defense, energy metabolism, age, genetics

## Abstract

**Simple Summary:**

The objective of this study was to measure the evolution with age of the antioxidant defense ability of two strains of chicken with fast or slow growth rates. Several parameters related to metabolic status, redox balance, and antioxidant defense activity were measured from hatching to 42 days of age, in liver, muscle (breast and thigh), and in plasma. Our results confirm the high level of oxidative stress just after hatching in chicks and, consequently, a high need for antioxidant defense during this period. Several enzymes or molecules, whose levels fall during the first week or two after hatching, would help to ensure this need. The antioxidant pathways used to maintain redox balance change with age and may differ between slow- and fast-growing chickens. SOD activity appears to be a key player in the antioxidant response of slow-growing chickens, while uric acid is thought to play a more important role in fast-growing chickens, particularly at the end of rearing.

**Abstract:**

The evolution of parameters known to be relevant indicators of energy status, oxidative stress, and antioxidant defense in chickens was followed. These parameters were measured weekly from 1 to 42 days in plasma and/or muscles and liver of two strains differing in growth rate. At 1-day old, in plasma, slow-growing (SG) chicks were characterized by a high total antioxidant status (TAS), probably related to higher superoxide dismutase (SOD) activity and uric acid levels compared to fast-growing (FG) chicks whereas the lipid peroxidation levels were higher in the liver and muscles of SG day-old chicks. Irrespective of the genotype, the plasma glutathione reductase (GR) and peroxidase (GPx) activities and levels of hydroperoxides and α- and γ-tocopherols decreased rapidly post-hatch. In the muscles, lipid peroxidation also decreased rapidly after hatching as well as catalase, GR, and GPx activities, while the SOD activity increased. In the liver, the TAS was relatively stable the first week after hatching while the value of thio-barbituric acid reactive substances (TBARS) and GR activity increased and GPx and catalase activities decreased. Our study revealed the strain specificities regarding the antioxidant systems used to maintain their redox balance over the life course. Nevertheless, the age had a much higher impact than strain on the antioxidant ability of the chickens.

## 1. Introduction

In poultry farms, birds face different types of stress, whether they are environmental (cold, heat, ventilation), technological (prolonged storage of eggs before hatching, delay for placing chicks after hatching, high rearing density), nutritional (dietary transitions, mycotoxins, acid oil, toxic metals, insufficient trace elements), or related to their health or immune status (pathologies, vaccination process) [1]. Modern strains of broilers that have been selected for high growth rate and breast meat yield are more susceptible to oxidative stress [2,3]. In comparison, the strains used for free-range Label Rouge production are characterized by slower growth rates, lower breast muscle yield [4], and are known to be more robust [5]. As mammals, birds are equipped with antioxidant defense systems (enzymes, glutathione, vitamins, mineral, etc.) that allow them to maintain the redox balance at the cellular and tissue levels [1]. In poultry, the development of these systems during embryo development has been well-described [6,7,8], but their evolution after hatching is not so clear, even though several studies have focused on this period. Indeed, Yang et al. [9] measured different markers related to the redox balance in the plasma of chickens fasted for 12 h at the ages of 14, 21, and 28 days but did not evidence any evolution of these markers with age. However, it has been shown that the antioxidant potential decreased significantly over the first 10 days post-hatching and was offset by an increase in glutathione peroxidase (GPx) activity in the liver [10]. Surai et al. [11] also observed a sharp decrease in hepatic vitamin E concentration in different avian species (chicken, turkey, duck and goose) during the first two weeks of post-hatch growth. Between 21 and 42 days, Del Vesco et al. [12] showed that in the liver, the glutathione (GSH) concentration decreased and catalase activity increased while the TBARS value (thio-barbituric acid reactive substances), a lipid peroxidation marker, and the GPx activity remained unchanged. Mahmoud and Edens [13] measured various forms of glutathione (oxidized and reduced, GSSG and GSH) in chicken blood and the activities of GPx and glutathione reductase (GR) at 2, 3, and 4 weeks of age. The GSH concentration and GPx activity decreased with age while the GR activity increased. Mizuno [14] assayed in the breast muscle of chickens Cu–Zn superoxide dismutase, Mn superoxide dismutase, catalase, glutathione peroxidase, and glutathione reductase activities and thio-barbituric acid-reactive products 1, 2, and 4 weeks, and 4 months after hatching. All of these enzyme activities declined as the chickens grew older. The aim of our study was to provide a general overview of the evolution during rearing of metabolic and redox status as well as antioxidant defense in chickens used for meat production. The evolution of several plasma and tissue markers was then monitored on a weekly basis during the first weeks of growth in two types of strains: a fast-growing (FG) strain used for standard production and a slow-growing (SG) strain used for free-range French Label Rouge or organic production. In this study, a large set of 15 metabolites, vitamins, or enzyme activities known as good indicators of the energy status, oxidative stress, and antioxidant defense of chickens were measured in plasma or/and in muscles and liver.

## 2. Materials and Methods

### 2.1. Animals and Experimental Design

All experimental procedures were performed in accordance with the French National Guidelines for the care and use of animals for research purposes (Certificate of Authorization to Experiment on Living Animals no. 7740, Ministry of Agriculture and Fish Products). The animal experiment was ethically approved by the French authorities under number APAFIS #22060-2019091916239724v2. Fast-growing (FG) Ross 308 and slow-growing (SG) JA 657 male chicks were furnished by Boyé Accouvage (La Boissière-en-Gâtine, France) and reared on wood straw under controlled conditions at the poultry experimental unit (PEAT) of INRAE Nouzilly (France). One hundred and twenty chicks per strain were distributed into two contiguous rooms (one room per strain) in a closed building to limit environmental disturbance and presenting the same environmental conditions (temperature, humidity, light duration, and intensity). The rearing density was 10 birds/m^2^ at the beginning of the experiment. All birds were individually identified by wing-banded and ad libitum fed diets under pelleted form. There were three feeding periods: starting (1–14 days), growing (15–28 days), and finishing (29–42 days). The composition and main characteristics of the diets are presented in Table 1. Diets were formulated according to the breeders’ recommendations. The vitamin E content of diets was 20 ppm (amount sufficient to cover growth needs) to avoid interfering with the antioxidant defenses of the chicks. To evaluate the growth performance, all chickens were weighed at D1, D14, D28, and D42 and the feed consumption per strain was registered for each period. To evaluate redox status, twenty-one-day-old chicks of each strain, chosen at random, were weighed and then sacrificed by cervical dislocation, alternating strains between each chick. Before sacrifice, blood samples were taken from the occipital sinus to recover plasma after centrifugation and stored at −80 °C. Samples of *Pectoralis major* muscle, a mix of thigh muscles, and liver were also collected, immediately frozen in liquid nitrogen, and stored at −80 °C. At 7, 14, 21, 28, 35, and 42 days of age, 10 birds per strain chosen at random were weighed, sacrificed, and sampled as at hatching. Eight hours before sacrifice, the animals were fasted but continued to have ad libitum access to water.

### 2.2. Determination of Plasma Metabolites and Redox Status of Chickens

The activities of superoxide dismutase (SOD), glutathione peroxidase (GPx), glutathione reductase (GR) and peroxidase, the total antioxidant status (TAS) and the concentrations of hydroperoxides (HPO), glucose, β-hydroxybutyrate (β-OH), free fatty acids (FFA), uric acid, and triglycerides were determined on plasma using a biological analysis robot (Arena 20XT, Thermo Scientific, Ilkirch, France) and commercial kits (Table 2). The TBARS index was determined according to Lynch and Frei [15] to estimate the lipid peroxidation. The concentrations of α- and γ-tocopherol were determined using a HPLC Thermo U3000 (Thermo Fisher Scientific, Illkirch, France) after the extraction of the organic phase with hexane (Faculty of Medicine, Marseille, France). On tissues, only the activities of SOD, GPx, GR, and catalase, and the TAS and the TBARS values were measured using the same methodologies as for plasma. All measurements of enzymatic activities were realized after the solubilization of around 100 mg tissue with 800 µL of a buffer containing 0.05 M Tris-HCl, 1 mM EDTA, and 0.25 M sucrose (pH 7.4). The tissue was first ground with a ball mill (Retsch MM400, Grosseron, Coueron, France) for 1 min, then centrifuged (10,000× *g*, 30 min, 4 °C), and the supernatant was collected and stored at −80 °C. For each tissue sample, the protein content was determined by UV–Visible spectrometry using a BCA Protein Assay Kit (Sigma-Aldrich, Saint-Quentin Fallavier, France) to express results relative to the protein contents.

### 2.3. Statistical Analysis

For all of the measured parameters, the replicate was the chicken. For the whole rearing period, the effects of age, strain, and their interaction were tested by analysis of variance using the Statview software (version 5.0) and a significance level at *p* ≤ 0.05.

## 3. Results

### 3.1. Growth Performance of Chickens

The body weight of chickens increased regularly with age (*p* < 0.001) from 51.6 g at D1 to 3134 g at D42 for the FG strain, and from 33.9 g at D1 to 1090 g at D42 for the SG strain (Figure 1). The effect of strain was significant (SG < FG, *p* < 0.001) for the whole experimental period and the difference between strains strongly increased with age. The growth performance of each group is presented in Table 3.

### 3.2. Plasma Metabolites

The glucose level remained quite stable between D1 and D42 in the FG strain. In contrast, it increased regularly from 2026 mg/L at D1 to 2577 mg/L at D28 and then decreased to 2362 mg/L at D42 in the SG strain (*p* < 0.001; Figure 1). The SG strain had a lower glycemia at D1 (2026 vs. 2342 mg/L, *p* < 0.05) but a higher one at D21 (2506 vs. 2186 mg/L, *p* < 0.05) than the FG strain.

The uric acid level of the SG strain decreased regularly from 87.44 mg/L at D1 to 36.66 mg/L at D21 (*p* < 0.001) and then remained stable (Figure 1). Interestingly, the opposite evolution was observed in the FG strain with an increase from 29.41 mg/L at D1 to 91.78 at D21 followed by a decrease to 31.31 mg/L at D35 and a further increase to reach 52.66 mg/L at D42 (*p* < 0.001). Consequently, the SG chickens were characterized by a higher uric acid content at D1 but lower contents at D14, D21, and D28 than the FG chickens (*p* < 0.05).

The triglyceride level decreased from 1061 or 462 mg/L at D1 to 266 or 135 mg/L at D7 in the FG and SG strains, respectively (*p* < 0.001; Figure 1) and then remained stable until D42. The FG strain had higher values at D1, D7, and D14 than the SG strain (*p* < 0.05), with the difference between the strains being particularly high at D1 when the FG chickens had 3-fold higher plasma triglyceride levels than the SG chickens.

The FFA level increased regularly from 217 µmol/L at D1 to 877 µmol/L at D21 and then remained stable in the FG strain (*p* < 0.01; Figure 1). In the SG strain, the FFA content increased from 96 µmol/L at D1 to 643 µmol/L at D7, followed by a decrease to 437 µmol/L at D14 and a further increase to 664 µmol/L at D42 (*p* < 0.01). Higher values in the FFA levels were observed in the plasma of FG chickens at all ages except for D7 and D14 (*p* < 0.05).

Finally, the β-OH level increased from 450 or 368 µmol/L at D1 to 737 or 821 µmol/L at D14 in the FG and SG strains, respectively (*p* < 0.05; Figure 1). Then, it remained stable except in the FG chickens for which we observed a strong decrease between D28 and D35. The SG strain had a higher β-OH content than the FG strain at this age (817 vs. 314 µmol/L, *p* < 0.001).

### 3.3. Plasma Redox Status and Enzyme Activities

The TBARS value was not affected by age and strain, except at D28 when the FG strain showed a higher value than the SG strain (*p* < 0.05; Figure 2).

The TAS value was differently affected by age according to the strain. For the FG strain, it increased from 1.04 mmol/L at D1 to 1.31 mmol/L at D28, then decreased to 0.93 mmol/L at D35 and increased again to 1.25 mmol/L at D42 (*p* < 0.05; Figure 2). In contrast, the TAS value decreased from 1.48 mmol/L at D1 to 0.88 mmol/L at D21, then remained stable until D42 in the SG strain. Regarding the strain effect, a lower TAS value was observed at D1 but higher values were observed at D21 and D28 in the FG chickens compared to SG chickens (*p* < 0.05).

The SOD activity increased regularly from 16.48 U/mL at D1 to 33.14 U/mL at D35, then remained stable in the FG strain (*p* < 0.01; Figure 2). However, it remained stable around 29–34 U/mL during the whole rearing period in the SG strain. The SOD activity remained lower from D1 to D21 in the FG strain before reaching similar levels than the SG strain (*p* < 0.01).

The evolution of the GPx activity with age is quite complex. It decreased between D1 and D7 from 8932 or 9198 U/L to 8226 or 8577 U/L in the FG and SG strains, respectively (*p* < 0.01; Figure 2). Then, it increased to 9417 or 9237 U/L at D14 in the FG and SG strains, respectively (*p* < 0.01), and remained stable in the SG strain while it decreased to 8205 U/L at D42 in the FG strain. There was no main effect of strain on this parameter (*p* > 0.05), even though at D42, the FG strain had a lower activity than the SG strain.

The GR activity decreased between D1 and D7 from 0.042 or 0.043 U/L to 0.028 or 0.025 U/L in the FG and SG strains, respectively (*p* < 0.001; Figure 2). Then, it remained stable in the SG strain while it increased to 0.048 U/L between D28 and D42 in the FG strain. As a consequence, GR activity was higher in the FG strain than in the SG strain at D35 and D42 (*p* < 0.05).

The peroxidase activity decreased regularly during the first two weeks of age from 3.68 or 4.689 mU/mL at D1 to 2.22 or 2.56 mU/mL at D14 in the FG and SG strains, respectively (*p* < 0.001; Figure 2). Then, it remained stable until D28 before increasing to 2.25 or 2.24 mU/mL at D35 and decreased to 1.14 or 1.37 mU/mL at D42 in the FG and SG strains, respectively (*p* < 0.001). The strain had no effect on this parameter (*p* > 0.05).

The hydroperoxide level decreased sharply from 11.67 or 16.48 µmol/mL at D1 to 2.22 or 2.56 µmol/mL at D14 for the FG and SG strains (*p* < 0.001; Figure 3). Then, it slightly increased at D21 to 5.19 or 4.62 µmol/mL and remained stable until D42 in the FG and SG strains, respectively (*p* < 0.001). The strain had no effect on this parameter (*p* > 0.05).

The α-tocopherol level sharply decreased during the two first weeks of age, from 53.99 or 34.15 ng/µL at D1 to 6.98 or 6.12 ng/µL at D14 in the FG and SG strains, respectively (*p* < 0.001). Then, it remained stable until D42 in the two strains. The α-tocopherol content at D1 was higher in the FG than in the SG strain (*p* < 0.05).

For the FG strain, the γ-tocopherol level decreased from 6.41 ng/µL at D1 to 2.06 ng/µL at D7 (*p* < 0.05; Figure 3). Then, it increased to 4.31 ng/µL at D14 and remained stable until D42. In the SG strain, the γ-tocopherol content decreased from 3.27 ng/µL at D1 to 1.69 ng/µL at D21 and then remained stable until D42 (*p* < 0.05). Differences between strains were observed at some ages, with higher values of γ-tocopherol observed in the FG strain than in the SG strain at D1, D14, and D35 (*p* < 0.05).

### 3.4. Muscles and Liver Lipid Peroxidation Status

In the liver, the TBARS value increased during the first week of age from 1.17 or 1.88 mg equivalent MDA/kg at D1 to 2.77 or 3.08 mg equivalent MDA/kg at D7 in the FG and SG strains, respectively (*p* < 0.001; Figure 4). Then, it decreased with age until reaching 1.53 or 1.45 mg equivalent MDA/kg at D42 in the FG and SG strains, respectively. Lower and higher values were observed in the FG chickens at D1 and D14, respectively, compared with the SG chickens.

In the breast muscle, the TBARS value sharply decreased during the first week, from 0.68 or 1.03 mg equivalent MDA/kg at D1 to 0.20 or 0.21 mg equivalent MDA/kg at D7 in the FG and SG strains, respectively (*p* < 0.001; Figure 4). Then, it remained stable in the SG strain until D42. In the FG strain, we observed a slight increase to 0.36 mg equivalent MDA/kg at D14 followed by a decrease between D21 and D28 and a stabilization thereafter. The FG strain had a lower value at D1 and D35 and a higher value at D21 than the SG strain (*p* < 0.05).

In the thigh muscles, the TBARS value decreased from 0.51 mg equivalent MDA/kg at D1 to 0.34 mg equivalent MDA/kg at D7 only in the FG strain (*p* < 0.05; Figure 4). Then, it remained stable until D42. In the SG strain, the TBARS value decreased later (between D7 and D14) from 1.02 mg equivalent MDA/kg to 0.65 mg equivalent MDA/kg (*p* < 0.001), then remained stable until D42. TBARS was much greater in the FG strain than in the SG strain between D1 and D21 (*p* < 0.01) before reaching similar values.

### 3.5. Muscles and Liver Antioxidant Status and Enzyme Activities

In the liver, the TAS value increased between D14 and D35, from 10.04 mmol/µg proteins to 18.56 mmol/µg proteins in the SG strain (*p* < 0.001; Figure 4). In the FG strain, the TAS value also increased between D14 and D28, from 8.99 mmol/µg proteins to 16.79 mmol/µg proteins, but decreased to 14.35 mmol/µg proteins at D35 and increased again at 16.54 mmol/µg proteins at D42 (*p* < 0.001). The FG strain had lower TAS values than the SG strain at D1, D7, and D35 (*p* < 0.05).

In the breast muscle, the TAS value increased from 0.296 mmol/µg proteins at D1 to 0.814 mmol/µg proteins at D42 in the SG strain (*p* < 0.001). In the FG strain, the TAS value increased from 0.457 mmol/µg proteins at D1 to 0.649 mmol/µg proteins at D21, but like in the liver, decreased to 0.487 mmol/µg proteins at D35 and increased again to 0.533 mmol/µg proteins at D42 (*p* < 0.001; Figure 4). The FG strain had higher values at D1 and D21 and lower values at D35 and D42 than the SG strain (*p* < 0.05).

In the thigh muscles, the TAS value decreased regularly with age from 0.550 mmol/µg proteins at D1 to 0.390 mmol/µg proteins at D42 in the SG strain (*p* < 0.001; Figure 4). In the FG strain, it increased from 0.482 mmol/µg proteins to 0.738 mmol/µg proteins at D7, then remained stable until D35 and decreased to 0.655 mmol/µg proteins at D42 (*p* < 0.001). From D7 to D42, the FG strain had much higher TAS values than the SG strain (*p* < 0.001).

In the liver, the GPx activity decreased from 4748 U/µg proteins at D1 to 4139 U/µg proteins at D7 and then remained stable until D42 in the FG strain (*p* < 0.01; Figure 5). It strongly decreased from 4612 U/µg proteins at D1 to 1712 U/µg proteins at D14, then increased to 4796 U/µg proteins until D28 and remained stable thereafter (*p* < 0.01) in the SG strain. The FG strain had higher values at D7 and D14, but a lower one at D28 than the SG strain (*p* < 0.05).

In the breast muscle, the GPx activity decreased from 1630 U/µg proteins at D1 to 1257 U/µg proteins at D14 in the FG strain (*p* < 0.01; Figure 5). It then remained quite stable until D35 before increasing at 1467 U/µg proteins at D42 (*p* < 0.01). In the SG strain, the GPx activity remained stable between D1 and D7, then decreased from 1438 U/µg proteins at D7 to 1268 U/µg proteins at D14. It increased to 1421 U/µg proteins at D28 and remained stable until D42 (*p* < 0.01). The FG strain had a higher value at D1 (*p* < 0.0001) and a lower value at D28 (*p* < 0.05) than the SG strain.

In the thigh muscles, the GPx activity decreased from 1557 or 1457 U/µg proteins at D1 to 1185 or 1206 U/µg proteins at D14 in the FG and SG strains, respectively (*p* < 0.001; Figure 5). Then, it increased regularly with age until 1357 or 1341 U/µg proteins at D42 for the FG and SG strains, respectively (*p* < 0.001). Like in the breast muscle, the FG strain had a higher value of GPx activity at D1 and a lower value at D28 (*p* < 0.05) than the SG strain.

In the liver, the evolution of the GR activity was similar between the two strains (Figure 5). It gradually increased from 0.796 or 0.717 U/µg proteins at D1 to 1.133 or 1.143 U/µg proteins for FG and SG, respectively, then remained quite stable until D42. The GR activity was higher at D28 in the SG strain compared with the FG strain (*p* < 0.05).

In the breast muscle, the GR activity decreased from 0.216 U/µg proteins at D1 to 0.127 U/µg proteins at D7 in the FG strain and from 0.157 U/µg proteins at D1 to 0.100 U/µg proteins at D14 in the SG strain, then remained quite stable until D42 for both strains (Figure 5). There was a strain effect on this parameter, with higher GR activity values observed in the FG strain at D1 and between D21 and D42 (*p* < 0.05).

In the thigh muscles, the GR activity evolved similarly than in the breast muscle (Figure 5). It decreased from 0.156 U/µg proteins at D1 to 0.108 U/µg proteins at D7 in the FG strain and from 0.140 U/µg proteins at D1 to 0.090 U/µg proteins at D14 in the SG strain, then remained quite stable until D42 in both strains. For the whole rearing period, the FG strain had higher values for GR activity than the SG strain (*p* < 0.01).

In the liver, the SOD activity was stable from D1 to D28 in the FG strain (*p* > 0.05, Figure 6), then increased from 126 to 268 U/µg proteins at D35 and decreased again to 159 U/µg proteins at D42 (*p* < 0.05). In the SG strain, the SOD activity increased from 679 U/µg proteins at D1 to 961 U/µg proteins at D14 (*p* < 0.001), then sharply decreased to 372 U/µg proteins at D21 and to 158 U/µg proteins at D42 (*p* < 0.001). The liver SOD activity was much lower in the FG strain than in the SG strain between D1 and D14 (*p* < 0.05).

In the breast muscle, the SOD activity increased from 20.41 or 19.30 U/µg proteins at D1 to 46.24 or 44.74 U/µg proteins at D7 in the FG and SG strains, respectively (*p* < 0.01, Figure 6). It then remained stable until D42 in the SG strain and decreased regularly with age to 27.25 U/µg proteins at D42 in the FG strain (*p* < 0.01). The FG strain had lower values than the SG strain from D14 to D42 (*p* < 0.01).

In the thigh muscles, the SOD activity increased from 4.15 U/µg proteins at D1 to 10.48 U/µg proteins at D7 in the FG strain (*p* < 0.001, Figure 6). Then, it remained stable until D21, decreased to 6.07 U/µg proteins at D28, and remained stable until D42 (*p* < 0.001). In the SG strain, the SOD activity strongly increased from 6.54 U/µg proteins at D1 to 28.30 U/µg proteins at D21 and then remained stable until D42 (*p* < 0.001). The FG strain had much lower values than the SG strain from D7 to D42 (*p* < 0.001).

In the liver, the catalase activity was not affected by age in the SG strain (*p* > 0.05). In the FG strain, decreased from 71.06 U/µg proteins at D1 to 44.23 U/µg proteins at D7 (*p* < 0.05) and then remained stable until D42. The FG strain had a higher catalase activity between D1 and D14 and a lower value at D21 than the SG strain (*p* < 0.05).

In the breast muscle, the catalase activity strongly decreased from 9.50 or 9.55 U/µg proteins at D1 to 0.929 or 1.257 U/µg proteins at D7 in the FG and SG strains, respectively (*p* < 0.001, Figure 6). Then, it remained stable until D42 in the two strains. The strain had no effect on this parameter (*p* > 0.05).

In the thigh muscles, the catalase activity decreased from 4.24 U/µg proteins at D1 to 2.79 U/µg proteins at D28 and then remained stable until D42 in the SG strain (*p* < 0.001, Figure 6). In the FG strain, it decreased from 7.81 U/µg proteins at D1 to 1.27 U/µg proteins at D28 and then remained stable until D42 (*p* < 0.001). The FG strain had higher values at D1 and D7 and lower values at D21 and D28 than the SG strain (*p* < 0.05).

## 4. Discussion

### 4.1. Effect of Age on the Markers of Redox Status and Plasma Metabolites of Chickens

For each sampling date, the chicks had been fasted for 8 h before sacrifice. The metabolites measured in plasma therefore reflected the basal metabolism of the animals. At D1, the plasma concentration in triglycerides was high but decreased rapidly during the first week where an increase in plasma FFA levels was observed. This can reflect the use by embryo and hatched chicks of lipids stored in the egg yolk and then in their liver as a major source of energy [16]. Moreover, β-hydroxybutyrate (β-OH), one of the main ketone bodies produced during the breakdown of fats in the body [17], strongly increased between D1 and D14. Moran et al. [18] and Uni and Ferket [19] have reported a high level of lipid peroxidation in the early stage after chick hatch. Lipid peroxidation, assessed through the TBARS index, was indeed maximal at D1 in both the breast and thigh muscles and D7 in the liver. The plasma concentration of hydroperoxides was also maximal at D1, then decreased until D14, confirming the high level of oxidative metabolism just after hatch. Consistently, the plasma levels of several enzymes or molecules involved in the antioxidant response declined within one to two weeks after hatching. This is the case for peroxidase, whose role is to break down toxic peroxides produced by oxidative stress [20], but also glutathione peroxidase (GPx) and reductase (GR) and α- and γ-tocopherol.

There are few data in the literature on the post-hatching evolution of poultry defenses against oxidative stress, and they are sometimes inconsistent. Yang et al. [9] measured the serum glutathione content, glutathione peroxidase activity, and total antioxidant in the serum of fasting chickens for 12 h at 14, 21, and 28 days of age but failed to show any significant variations with age. However, between 3 and 6 weeks of age, the hepatic glutathione (GSH) concentration decreased and the catalase activity increased while the TBARS value and glutathione peroxidase activity were unchanged [12]. In our study, the catalase activity remained quite stable in the liver after 7 days. By measuring the various forms of glutathione (oxidized and reduced, GSSG and GSH) and activities of the glutathione peroxidase and reductase in chicken blood, Mahmoud and Edens [13] showed that GSH concentration and GPx activity decreased between 2 and 4 weeks while GR activity increased. Again, these last results differ from ours, since we observed stable GR and GPx activities between 2 and 4 weeks in both strains. Such discrepancies between studies highlight the fact that the evolution of antioxidant status in chickens is probably influenced by many factors related to animal genetics, nutrition, or the management of farming conditions.

From our results, it is interesting to note that the high oxidative metabolism measured in chicks the first week after hatching does not imply a decrease in the total antioxidant status either in plasma but also in the liver and muscles. This could be explained by the use of vitamin E (tocopherol), a strong antioxidant molecule whose concentration in plasma strongly decreased during the first week post-hatching. This is consistent with previous results that showed a sharp decrease in hepatic vitamin E concentration in different avian species (chicken, turkey, duck, and goose) during the first two weeks of post-hatch growth [11].

Just after hatch, the main enzymes acting as antioxidants seem to be the peroxidase and glutathione reductase, whose activities decreased thereafter in plasma but also in the liver and muscles. Catalase activity was also maximal one day after hatch in liver and muscles, also suggesting a major role of this enzyme in maintaining the redox balance of chicks at this stage. In the breast muscle, the catalase activity decreased 10-fold during the first week and remained stable at low-levels thereafter, suggesting a minor role of this enzyme in the antioxidant response of this tissue in the later stages. SOD then appears to take over the antioxidant defense of chickens at the plasma and tissue levels in both strains.

### 4.2. Effect of Strain on the Markers of Redox Status and Plasma Metabolites of Chickens

From D1 to D14, the FG strain had a higher plasma triglyceride content than the SG strain. At D1 and after D21, the plasma concentration in FFA was lower for the SG strain compared to the HG strain, suggesting a lower lipolysis activity or a higher lipid oxidation. Indeed, at D1 in the liver and breast muscle, and from D1 until D21 in the thigh muscles, the lipid peroxidation evaluated with the TBARS value was higher for the SG chicks compared to the FG chicks. This would be more related to the energetic metabolism of SG chicken muscles, which have a higher oxidative activity and a higher iron content (myoglobin and hemoglobin) in muscles than FG chickens because of their greater physical activity [21,22]. Castellini et al. [23] also reported higher lipid peroxidation estimated with the TBARS value in the leg muscles of chickens with a slow growth rate than in the leg muscles of chickens with a high growth rate. However, in the plasma, the strain had no effect on the lipid peroxidation, the hydroperoxide concentration, and the peroxidase activity. Moreover, at D1, the plasma concentration in uric acid was higher in SG chicks compared to FG chicks, suggesting a higher catabolism of proteins. The chicks of the two strains were received on the same day, but it is possible that those of the SG strain hatched earlier than those of the FG strain, hence a greater exhaustion of their energy reserves and the obligation to increase their protein catabolism. This could also explain the difference at D1 between strains on the α- and γ-tocopherol contents in plasma. Another possibility is a different supplementation with tocopherol in the diets of breeders producing SG and FG chicks. At D14, D21, and D28, the FG strain had a higher plasma concentration in uric acid than the SG strain, suggesting a higher protein turnover, probably to allow for the high development of breast muscles. The strain had few effects on the GPx activity in the plasma and muscles. In the liver, this activity was lower at D7 and D14 and higher at D28 for the SG strain compared to the FG strain [24]. The strain had few effects on the GR activity in the liver and thigh muscles. However, the FG strain had a higher GR activity than the SG strain in the breast muscle and plasma at D35 and D42. The FG strain had a lower SOD activity in the plasma from D1 to D21, in the liver from D1 to D14, in the thigh muscle from D7 to D42, and in the breast muscle from D14 to D42 than the SG strain, which had a higher locomotor activity and higher oxidative metabolism in muscles. Whatever the tissue considered, the SOD activity seemed to differentiate the two strains. For the catalase activity, the FG strain had higher values than the SG strain in the liver from D1 to D14 and in the thigh muscles at D1 and D7 whereas the strain had no effect on the catalase activity in the breast muscle. The SG strain had a higher TAS value than the FG strain at D1 in the plasma, at D1, D7, and D35 in the liver, and at D35 and D42 in the breast muscle, whereas the FG strain had a higher TAS value than the SG strain at D1 and D21 in the breast muscle, and from D7 to D42 in the thigh muscles, which may be in relation to a higher locomotor activity and a higher oxidative metabolism in SG chicks. In the plasma, the evolution with age of TAS value for each strain seemed to be related to that of uric acid content, this metabolite also having antioxidant properties [25]. The activity of enzymes implicated in the antioxidant defense is different according to the strain and the tissue considered. Finally, the strain had few effects on the plasma concentration in glucose and β-OH.

Differences observed between strains suggest that fast-growing broilers might use different pathways than slow-growing birds to maintain their redox balance during growth. Catalase, GPx, and GR would be preferentially mobilized just after hatching in the liver or muscles in both strains, whereas GR in the pectoral muscle and uric acid in plasma would take over later in development to provide antioxidant defense in FG birds, as also suggested by Machin et al. [26]. In contrast, the SOD pathway, whose activity remained at high levels over time in the blood and tissues, could play a major role in the SG chickens. Finally, the increase in plasma uric acid content specifically observed between hatch and three weeks of age in the FG broilers may also reflect their high protein turnover, allowing them to maintain their high growth rate. By exploring oxidative phosphorylation in breast muscle, Hubert and Athrey [27] found that it was significantly reduced in FG broilers compared to SG chickens.

## 5. Conclusions

Our results confirm the high level of oxidative stress just after hatching in chicks, and consequently, a high need for antioxidant defense during this period. Several enzymes or molecules, among which vitamin E, catalase, peroxidase, glutathione peroxidase, and reductase, whose levels drop significantly in the first one to two weeks post-hatching, would ensure this. Interestingly, our study revealed that during the first stage of growth, slow-growing chickens had a lower energy status linked to stronger oxidative metabolism and antioxidant response compared to fast-growing chickens. During growth, the antioxidant pathways used to maintain redox balance in chickens change, and may be different between slow and fast-growing genotypes. SOD activity seems to be a key player in the antioxidant response of slow-growing chickens, solicited first at the blood and hepatic levels and then at the muscle level, while uric acid could play a more important role in the antioxidant response of fast-growing chickens, especially in the late stages of development. The detailed description of the evolution of metabolic status and antioxidant defenses during the growth of two genotypes of chicken with slow and fast growth can be useful for the research of solutions adapted to the various contexts of production to maintain a good redox balance, and thus good health throughout the life of the animal.

## Figures and Tables

**Figure 1 animals-13-01160-f001:**
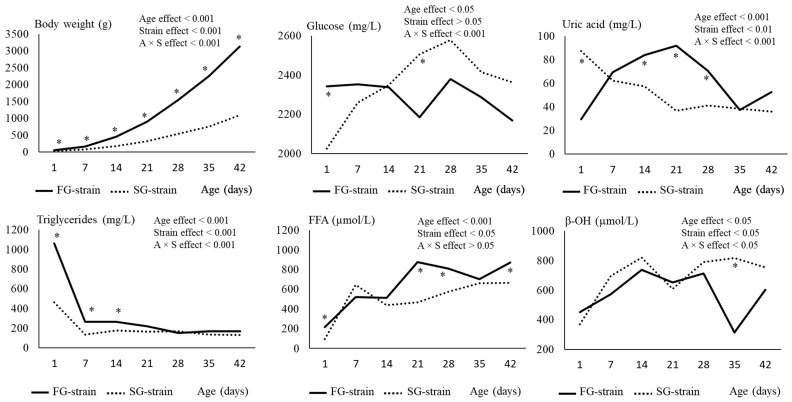
Effects of age (A) and strain (S) and their interaction on chicken body weight and plasma metabolite content. Data are presented as means; *n* = 10 per strain and per age. *: Significant difference (*p* < 0.05) between strains for a given age; FG = fast-growing, SG = slow-growing.

**Figure 2 animals-13-01160-f002:**
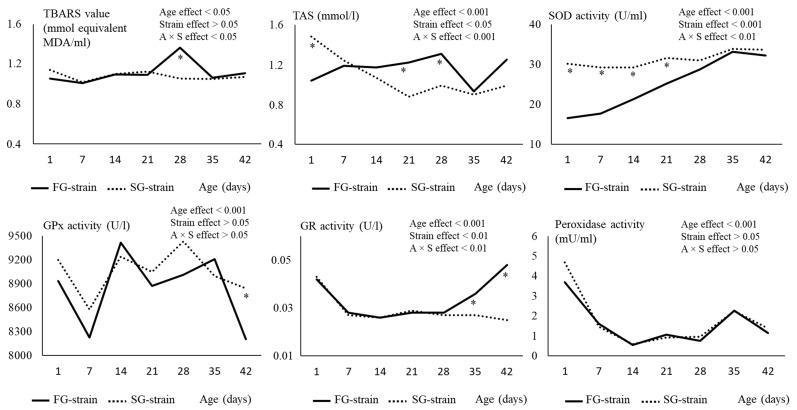
Effects of age (A) and strain (S) and their interaction on plasma TBARS, TAS and SOD, GPx, GR, and peroxidase activity. Data are presented as the means; *n* = 10 per strain and per age. *: Significant difference (*p* < 0.05) between strains for a given age; FG = fast-growing, SG = slow-growing.

**Figure 3 animals-13-01160-f003:**
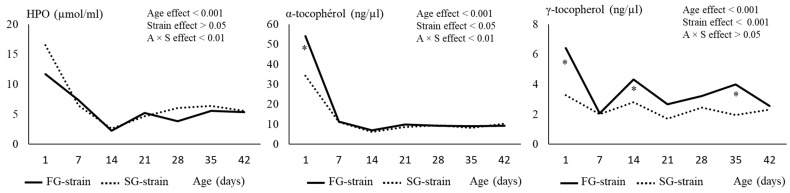
Effects of age (A) and strain (S) and their interaction on plasma hydroperoxides (HPO) and α- and γ-tocopherol content. Data are presented as the means; *n* = 10 per strain and per age. *: Significant difference (*p* < 0.05) between strains for a given age; FG = fast-growing, SG = slow-growing.

**Figure 4 animals-13-01160-f004:**
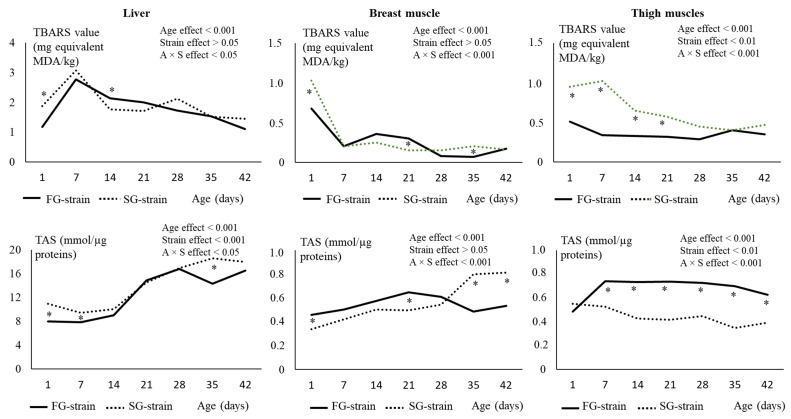
Effects of age (A) and strain (S) and their interaction on thio-barbituric reactive substance (TBARS) content and total antioxidant status (TAS) in the liver, breast and thigh muscles. Data are presented as the means; *n* = 10 per strain and per age. *: Significant difference (*p* < 0.05) between strains for a given age; FG = fast-growing, SG = slow-growing.

**Figure 5 animals-13-01160-f005:**
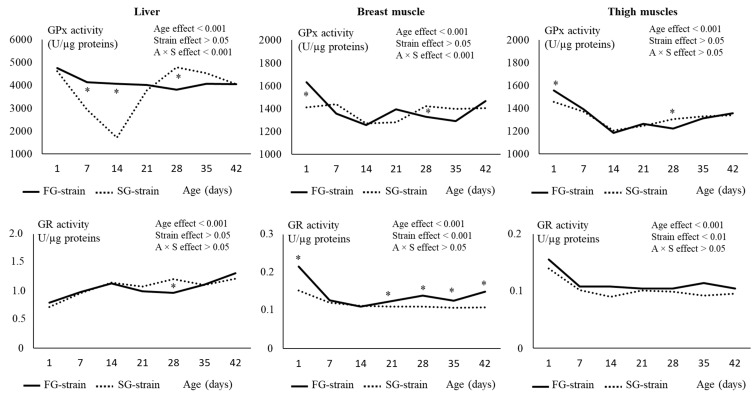
Effects of age (A) and strain (S) and their interaction on glutathione peroxidase (GPx) and glutathione reductase (GR) in the liver, breast, and thigh muscles. Data are presented as the means; *n* = 10 per strain and per age. *: Significant difference (*p* < 0.05) between strains for a given age; FG = fast-growing, SG = slow-growing.

**Figure 6 animals-13-01160-f006:**
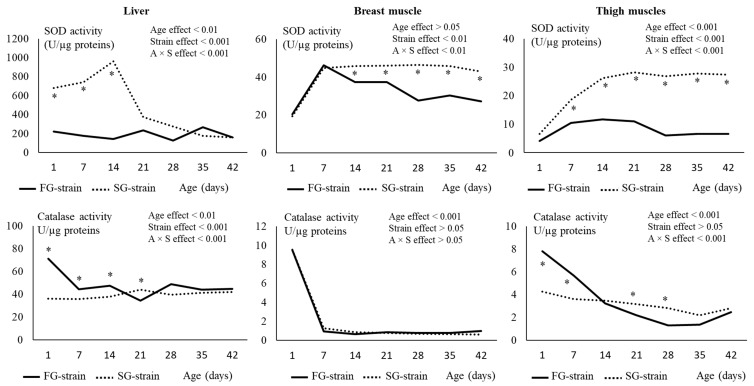
Effects of age (A) and strain (S) and their interaction on superoxide dismutase (SOD) and catalase in the liver, breast, and thigh muscles. Data are presented as the means; *n* = 10 per strain and per age. *: Significant difference (*p* < 0.05) between strains for a given age; FG = fast-growing, SG = slow-growing.

**Table 1 animals-13-01160-t001:** Composition and main calculated characteristics (g/kg) of diets.

	Standard Diets			Label Rouge Diets		
	1–14 d	15–28 d	29–42 d	1–14 d	15–28 d	29–42 d
Wheat	350.00	420.04	456.56	333.60	350.00	500.00
Soybean meal	342.61	100.00	100.00	334.83	100.00	100.00
Corn	237.32	163.11	200.00	268.24	343.95	200.00
Rapeseed meal		78.64	80.00		40.75	80.00
Dehulled sunflower meal		65.00	14.59		65.00	18.95
Corn gluten		70.00	40.00		16.27	
Tradigriller ^1^		30.00	50.00		30.00	50.00
Soya oil	18.11	16.90	3.70	15.47		5.25
Rapeseed oil	10.00	20.00	20.00	10.00	20.00	20.00
Dicalcium phosphate	19.36	14.18	13.57	18.88	14.20	9.05
Calcium carbonate	11.53	9.32	10.27	9.76	7.52	6.07
Vitamins and trace minerals ^2^	4.0	4.0	4.0	4.0	4.0	4.0
NaCl	3.0	3.0	3.0	3.0	3.0	3.0
HCl-lysine	1.71	4.46	2.96		3.38	1.92
dl-methionine	1.62		0.48	1.72	1.02	1.26
l-threonine	0.24	0.41	0.15		0.28	
l-tryptophane		0.44	0.22		0.13	
Coccidiostat ^3^	0.50	0.50	0.50	0.50	0.50	0.50
Metabolizable energy MJ/kg	11.91	12.33	12.33	11.91	12.12	12.33
Crude protein	220	205	185	216	170	166
Lysine	12.50	11.20	9.70	10.99	9.44	8.65
Sulfur amino-acids	8.30	7.50	7.20	8.38	7.12	7.20
Methionine	4.66	3.61	3.57	4.75	3.86	3.87
Tryptophane	2.65	2.50	2.15	2.60	1.90	1.89
Threonine	8.10	7.50	6.60	7.79	6.19	5.82
Calcium	10.80	9.00	9.10	10.00	8.00	6.50
Available phosphorus	4.40	3.80	3.70	4.30	3.60	3.00

^1^ Tradigriller = extruded linseeds + faba beans (50/50) (Valorex, Combourtillé, France). ^2^ Vitamins and trace minerals: ascorbic acid 750 mg; cholecalciferol 21.5 mg; alpha-tocopherol acetate 5000 mg; menadione 1000 mg; thiamine 1000 mg; riboflavine 1600 mg; d-pantothenate calcium 5000 mg; chlorhydrate pyridoxine 1400 mg; cobalamine 5.2 mg; niacine/niacinamide 20,000 mg; folic acid 600 mg; biotin 60 mg; choline chloride 110,000 mg; copper 3200 mg; iron 10,000 mg; zinc 14,000 mg; manganese 16,000 mg; iodine 400 mg; selenium 40 mg. Values are expressed per kg of commercial product. ^3^ Coccidiostat = Maxiban (Elanco, Sèvres, France).

**Table 2 animals-13-01160-t002:** List of kits used for the determination of the plasma metabolites and redox status of chickens.

Kit Name	Suppliers	Reference	Postal Address of Suppliers
Gpx (glutathione peroxidase)	Randox	RS505	55 Diamond, Crumlin, County Antrim, BT29 4QY, UK
TAS (total antioxidant status)	Randox	NX2332	55 Diamond, Crumlin, County Antrim, BT29 4QY, UK
SOD (superoxide dismutase)	Sigma Aldrich	19160-1KT-F	Sigma Aldrich Chimie S.a.r.l 80 rue de Luzais, L’Isle d’Abeau Chesnes St Quentin Fallavier Cedex 38297, France
NEFA-HR2 (Free Fatty Acids)	Fujifilm	434-91795	FUJIFILM Wako Chemicals Europe GmbH, Fuggerstraße 12, 41468 Neuss, Germany
		436-91995	
Glucose (GOD-POD)	ThermoFisher	981780	ThermoFisher Scientific 2 avenue des Chaumes, 78180 Montigny le Bretonne, France
β-OH (β-hydroxybutyrate acid)	ThermoFisher	984392	ThermoFisher Scientific 2 avenue des Chaumes, 78180 Montigny le Bretonne, France
Triglycerides	ThermoFisher	981786	ThermoFisher Scientific 2 avenue des Chaumes, 78180 Montigny le Bretonne, France
GR (glutathione reductase)	Sigma Aldrich	GRSA-1KT	Sigma Aldrich Chimie S.a.r.l 80 rue de Luzais, L’Isle d’Abeau Chesnes St Quentin Fallavier Cédex 38297, France
Uric acid	ThermoFisher	981788	ThermoFisher Scientific 2 avenue des Chaumes, 78180 Montigny le Bretonne, France
Hydro-peroxides and peroxidase activity	Cell Biolabs	STA-844	Cell Biolabs 7758 Arjons Drive San Diego, CA 92126, USA
Catalase activity	ThermoFisher	EIACATC	ThermoFisher Scientific 2 avenue des Chaumes, 78180 Montigny le Bretonne, France

**Table 3 animals-13-01160-t003:** Growth performance of chickens from the fast (FG) and slow growth (SG) rate strains.

	FG Strain	SG Strain	Strain Effect
BW D1 (g)	45.18 ^a^ (*n* = 120)	33.30 ^b^ (*n* = 120)	<0.001
Feed consumption D1–D14 (g/d)	33.74 (*n* = 1)	21.29 (*n* = 1)	nd
DWG D1–D14 (g/d)	28.05 (*n* = 1)	10.53 (*n* = 1)	nd
FCR D1–D14	1.20 (*n* = 1)	2.02 (*n* = 1)	nd
BW D14 (g)	437.86 ^a^ (*n* = 89)	180.75 ^b^ (*n* = 87)	<0.001
Feed consumption D14–D28 (g/d)	116.55 (*n* = 1)	41.37 (*n* = 1)	nd
DWG D14–D28 (g/d)	78.06 (*n* = 1)	25.23 (*n* = 1)	nd
FCR D14–D28	1.49 (*n* = 1)	1.64 (*n* = 1)	nd
BW D28 (g)	1530.75 ^a^ (*n* = 78)	534.03 ^b^ (*n* = 77)	<0.001
Feed consumption D28–D42 (g/d)	192.15 (*n* = 1)	115.67 (*n* = 1)	nd
DWG D28–D42 (g/d)	114.55 (*n* = 1)	39.73 (*n* = 1)	nd
FCR D28–D42	1.67 (*n* = 1)	2.91 (*n* = 1)	nd
BW D42 (g)	3134.40 ^a^ (*n* = 57)	1090.20 ^b^ (*n* = 56)	<0.001

BW = body weight; DWG = daily weight gain; FCR = feed conversion ratio; nd = not determined. a, b: significant differences between strains with *p* < 0.001.

## Data Availability

Not applicable.

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
