# Peer review of "Slow and Fast-Growing Chickens Use Different Antioxidant Pathways to Maintain Their Redox Balance during Postnatal Growth"

_animals, 2023, doi:10.3390/ani13071160_

Round 1
Reviewer 1 Report
Dear authors,
I read your manuscript (animals-2279472) with interest and found much merit in your work. The manuscript, on the evolution of the redox balance in postnatal growth, is interesting and helps to understand the pathways of the redox balance. The paper fits the scope of the journal.
But some issues with the manuscript limit a publication in its current form.
Materials and Methods section
As it has been not described in this section, all the chicks of a same strain were distributed in a unique pen. This means that the FCR cannot be analyzed statistically. So, you cannot write, for example, “average feed consumption but a lower FCR than the SG strain” in lines 141 and 142.
Did you identify individually the chicks? It is not said. If you don’t, you cannot calculate the DWG individually and so, you cannot say n=50 (as you wrote in Table 3). Please clarify if you have identified the chicks or change the n: you would have only one data per strain.
Also, specify if you have weighed all the chicken at every age. In Table 3 you wrote different n-values in the BW (not n=21 for the 1st day or n=10 for the other).
The used feed seems to be appropriate, but the use of enzymes (such as xylanases) is not specified. Xylanases are necessary if you use near 40% of wheat in the diets. Please write it in the Table 1.
Also on Table 1. The values of Vitamins and trace minerals are expressed as per kg of feed or per kg of commercial product? Please clarify.
In the statistical analysis (line 132), use the expression “Analysis of variance” instead of “variance analysis”.
Results section
Data on lines 137 and 138 differ from those in Table 3. Please correct.
On Table 3, p-value of the Strain effect should be written <0.001 instead 0.001. Please correct.
Also on Table 3, g/j must be changed to g/d.
And also on Table 3, n-values of DWG D28-D42 and BW D42 don’t fit. Please correct.
Also on Table 3, please specify what “nd” means.
On line 162 it is better not to write “the strain had no effect per se”… “the S G strain had a lower glycaemia at D1...”. The interaction effect explains exactly this.
On line 175 write “3-fold” instead “4-fold”.
On line 258, please change “had no effect” by “had effect”.
The results section must be strongly reduced. Figures are ok and help to view the trends. But it is not useful to explain every single data in the text. A proposal could be to create Tables with the data, that would be easier to view and read.
You could also perform a linear and non-linear contrasts to check the trend of the changes with the age. With this type of analysis, you can know if the punctual differences in a single day are important or not. In this sense, when you find a significative interaction between age and strain, you could explain better the effect if you analyze the trend separately by strain.
Discussion
In this section, some information is repeated from Results section. For example, in every paragraph from line 399 you start rewriting the trends of the different parameters. This duplicate information must me removed from the section.
Moreover, the Discussion must be improved with stronger arguments supported by bibliographic references.
Author Response
First, I thank you very much for your comments and corrections that help me to improve my manuscript.
The paragraph concerning results on growth performance was modified. All chickens were individually identified but the DWG was calculated taking into account the average body weight of the group, so n = 1.
The evaluation of growth performance of each strain was clarified in the material and methods. Xylanases were not added to the diets. For minerals and vitamins, values are expressed per kg of commercial product. “Variance analysis” was replaced by “analysis of variance”.
Data on lines 137-138 were obtained with the birds sacrificed at D1 and D42, so 20 and 10 chicks, respectively.
Table 3 was corrected according your recommendations. Below table 3, the mention nd was clarified.
The mention “The strain had no effect per se” was suppressed in all paragraphs of results presentation.
We replaced “almost 4-fold” by “3-fold” as recommended.
On line 258, the sentence was modified.
As this study provided many data, we preferred to present them with figures instead of tables to help to view the trends with age and differences between strains. Only values related to significant differences between strains or ages are reported in the text. Indicating all data in tables will represent many pages.
The discussion was modified according to your recommendations and references were added.
Reviewer 2 Report
The authors investigated the antioxidant status of a slow and a fast growing strain weekly from hatching to slaughter age.
The manuscript is well written and the results are well presented.
I have only a few comments:
- What is the meaning of the abbreviation ‘j’ in table 3?
- Table 3: It would be clearer if the letters were superscripted that indicate a statistical difference.
- Printed in black and white, the colors green and orange are not well visible. A solid and a dashed line would be better suited here. Data points would make the diagrams even clearer. (Figure 1-6)
- Note line spacing in lines 377-381 and lines 505-520.
- I am missing a statement on why slow- and fast-growing broilers differ in their antioxidant status and how that specifically affects practical applications. Chronic stress has been reported to affect antioxidant enzymes. Fast growing broilers are exposed to higher physiological stress due to faster muscle growth. Could this have anything to do with the differences in antioxidant status? Is it possible to conclude from the results that one of the strains suffer from oxidative stress?
Author Response
First, I thank you very much for your comments and corrections that help me to improve my manuscript. In table 3, I forgot to translate the unit from French to English. I modified g/j to g/d. The letters to indicate statistical differences were superscripted as recommended. I also modified the figures to have visible differences between HG and SG strains when printed in black and white. The line spacing was corrected in lines 377-381 and lines 505-520. SG chickens have a more oxidative energy metabolism in muscles than HG chickens. This could explain the role of SOD more important for SG chickens. On the reverse, the protein metabolism and accretion in the muscles of HG chickens produce high amount of uric acid that could play a main antioxidant role for this strain. We cannot conclude that one of the strains suffers from oxidative stress as this implies that the antioxidant ability of birds has been overwhelmed and then tissue injuries appear inducing inflammatory reactions. In the present study, we did not explore these mechanisms and chickens were not exposed to stressful conditions.
Reviewer 3 Report
The work is very interesting and brings new elements regarding antioxidant reactions in the body of slow and fast growing broiler chickens.
The purpose of the work is clearly formulated. The material used for the research is sufficient, the research methods have been selected appropriately and described in detail. However, I lack information about the conditions of keeping the chickens: stocking density/m2. Whether the temperature and humidity in the rooms were at a similar level. If these parameters differed, it could be a factor that influenced the reactions of the chickens. If the authors have these data, they should be provided.
The results are presented in 3 tables and 6 figures. In table 3, it is not understandable to mark the significance of differences with the letters "a" and "c" next to BW D1, BW D14; BW D28. It should be "a" and "b" as for BW D42. In addition, DWG D1-D14; DWG D14-D28 and DWG D28-D42 also differ significantly between groups, so letters should be used.
The figures are legible, and the description in the text of the obtained results complements the data contained in the figures.
The discussion of the results against the background of the results of other authors is detailed. The publications cited by the authors of the article are properly selected. The conclusions are correct and result from the obtained research results.
Author Response
First, I thank you very much for your comments and corrections that help me to improve my manuscript. At the beginning of the study, the stocking density was 10 birds/m2 but decreased regularly with chicken age as 10 individuals per strain were sacrificed each week. The rearing conditions (temperature, humidity, light duration and intensity) were equivalent for the two strains. I added this information in the manuscript. In table 3, I corrected and added letters after the values significantly different.
Round 2
Reviewer 1 Report
Dear authors,
I have read the new version, and you have addressed most of the suggestions. It is not possible to solve the issue of not having used enzymes (xylanases), but it is not a crucial part of the paper. However, I would still recommend using them next time when preparing a diet with a significant proportion of wheat to avoid digestibility problems with chickens. In my opinion, the paper can be published in its current form.